# First-Line Pembrolizumab Mono- or Combination Therapy of Non-Small Cell Lung Cancer: Baseline Metabolic Biomarkers Predict Outcomes

**DOI:** 10.3390/cancers13236096

**Published:** 2021-12-03

**Authors:** David Lang, Linda Ritzberger, Vanessa Rambousek, Andreas Horner, Romana Wass, Kaveh Akbari, Bernhard Kaiser, Jürgen Kronbichler, Bernd Lamprecht, Michael Gabriel

**Affiliations:** 1Department of Pulmonology, Johannes Kepler University Hospital Linz, Krankenhausstraße 9, 4020 Linz, Austria; david.lang@kepleruniklinikum.at (D.L.); vanessa.rambousek@kepleruniklinikum.at (V.R.); andreas.horner@kepleruniklinikum.at (A.H.); romana.wass@kepleruniklinikum.at (R.W.); bernhardkaiser@gmx.at (B.K.); bernd.lamprecht@kepleruniklinikum.at (B.L.); 2Medical Faculty, Johannes Kepler University Linz, Altenberger Strasse 69, 4020 Linz, Austria; linda.ritzberger@gmx.at; 3Central Radiology Institute, Johannes Kepler University Hospital Linz, Krankenhausstraße 9, 4020 Linz, Austria; kaveh.akbari@kepleruniklinikum.at; 4Institute of Nuclear Medicine and Endocrinology, Johannes Kepler University Hospital Linz, Krankenhausstraße 9, 4020 Linz, Austria; juergen.kronbichler@kepleruniklinikum.at

**Keywords:** total metabolic tumor volume, bone marrow to liver ratio, PET/CT, overall survival, immunotherapy, immune checkpoint inhibitor, standardized uptake value, response prediction

## Abstract

**Simple Summary:**

Positron-emission tomography/computed tomography (PET/CT) is used for staging of non-small cell lung cancer (NSCLC) and can help to estimate prognosis in patients treated with immune checkpoint inhibitor (ICI) therapy. Most available data in that field were derived from cohorts treated in higher therapy lines using ICI monotherapy with different drugs. Currently, however, most advanced NSCLC patients receive first-line ICI treatment, often in combination with cytotoxic chemotherapy. We evaluated prognostic PET/CT biomarkers in 85 patients receiving first-line ICI, 70 (82%) of them as a chemotherapy–ICI combination. We found that patients with a higher metabolically active tumor volume (MTV) had a significantly poorer survival and lower radiological response rate. In patients with high MTV, a concomitantly low bone marrow to liver ratio indicated a better prognosis. Our results demonstrate that PET/CT-derived biomarkers can aid therapeutic decision-making in ICI-treated NSCLC.

**Abstract:**

Quantitative biomarkers derived from positron-emission tomography/computed tomography (PET/CT) have been suggested as prognostic variables in immune-checkpoint inhibitor (ICI) treated non-small cell lung cancer (NSCLC). As such, data for first-line ICI therapy and especially for chemotherapy–ICI combinations are still scarce, we retrospectively evaluated baseline ^18^F-FDG-PET/CT of 85 consecutive patients receiving first-line pembrolizumab with chemotherapy (*n* = 70) or as monotherapy (*n* = 15). Maximum and mean standardized uptake value, total metabolic tumor volume (MTV), total lesion glycolysis, bone marrow-/and spleen to liver ratio (BLR/SLR) were calculated. Kaplan–Meier analyses and Cox regression models were used to assess progression-free/overall survival (PFS/OS) and their determinant variables. Median follow-up was 12 months (M; 95% confidence interval 10–14). Multivariate selection for PFS/OS revealed MTV as most relevant PET/CT biomarker (*p* < 0.001). Median PFS/OS were significantly longer in patients with MTV ≤ 70 mL vs. >70 mL (PFS: 10 M (4–16) vs. 4 M (3–5), *p* = 0.001; OS: not reached vs. 10 M (5–15), *p* = 0.004). Disease control rate was 81% vs. 53% for MTV ≤/> 70 mL (*p* = 0.007). BLR ≤ 1.06 vs. >1.06 was associated with better outcomes (PFS: 8 M (4–13) vs. 4 M (3–6), *p* = 0.034; OS: 19 M (12-/) vs. 6 M (4–12), *p* = 0.005). In patients with MTV > 70 mL, concomitant BLR ≤ 1.06 indicated a better prognosis. Higher MTV is associated with inferior PFS/OS in first-line ICI-treated NSCLC, with BLR allowing additional risk stratification.

## 1. Introduction

Positron-emission tomography/computed tomography (PET/CT) is widely applied for staging both limited and advanced non-small cell lung cancer (NSCLC) [1,2,3,4]. In the last decade, immune-checkpoint inhibitors (ICI) directed against programmed cell death protein 1/programmed death-ligand 1 (PD-1/PD-L1) brought major therapeutic advances in NSCLC. Originally introduced as second-line therapy [5,6,7,8], PD-1/PD-L1-blockade alone, or as a combination together with platinum-based doublet chemotherapy has moved into first-line treatment, leading to improved survival and frequently to long-term responses [2,9,10,11,12,13]. The challenge of predicting favorable responses is still ongoing, whereas biomarkers such as PD-L1 expression, tumor mutational burden, or presence of targetable genetic tumor alterations are being widely applied [14]. Moreover, clinical or laboratory parameters such as Eastern Cooperative Oncology Group (ECOG) performance status, neutrophil to lymphocyte ratio, lactate dehydrogenase, or C-reactive protein (CRP) have been suggested [14,15,16], but each of them provides only limited prognostic properties on the individual patient’s level. Several biomarkers derived from PET/CT imaging have been reported to predict outcomes in various malignancies treated with ICI [4,17,18]. Concerning NSCLC treated with chemotherapy or ICI, especially volume-based PET/CT variables such as total (whole-body) metabolic tumor volume (MTV) and total lesion glycolysis (TLG) have been shown prognostic properties in terms of therapy response and survival [19,20,21,22,23,24,25,26,27]. Inflammatory processes within the tumor microenvironment are a major pathogenetic element in lung cancer [28], and biomarkers reflecting systemic inflammation are associated with reduced prognosis in lung cancer patients [14,16,29]. Of interest, the combination of the quantitative PET/CT biomarker MTV and the blood-based inflammation biomarker derived neutrophil to lymphocyte ratio (DNLR) had a prognostic impact in NSCLC patients receiving ICI [27,30,31,32]. Importantly, ^18^F-FDG not only accumulates in tumor cells but also in activated immune cells both in malignant as well as in inflammatory processes [17,33,34,35]. In cancer patients, FDG uptake represents not only immunological processes in the tumor microenvironment, but also allows an estimation of the activity of lymphatic tissues, as usually expressed by the bone marrow to liver ratio (BLR) or the spleen to liver ratio (SLR) [35,36]. In malignant melanoma, higher BLR as well as SLR have been reported to be associated with an unfavorable prognosis [37,38]. High bone marrow activity has also been suggested as a prognostic biomarker in gynecological cancers [39,40], and similar prognostic implications for high SLR could be shown for resected rectal or breast cancer [41,42]. Concerning NSCLC, bone marrow hypermetabolism has been reported as a prognostic factor after resection or in chemo(-radio)therapy settings [43,44,45,46]. Recently, high SLR has been reported as a significant predictor of reduced one-year PFS and two-year OS in first-line ICI monotherapy-treated advanced NSCLC patients [46].

Importantly, with some exceptions [22,25,26,46], most of the existing evidence for quantitative PET/CT biomarkers in the context of ICI therapy is based on mono-immunotherapy cohorts in higher therapy lines, reflecting the initial regulatory approvals for nivolumab, pembrolizumab, and atezolizumab [5,6,7,8]. However, first-line ICI therapy in combination with chemotherapy or as monotherapy for tumors with PD-L1 expression ≥50% is currently regarded as standard of care [1,2,10,11,47,48], Whether the existing data on quantitative PET/CT biomarkers can be transferred to the present therapeutic setting is thus questionable, especially due to the increased application of chemotherapy together with ICI.

Consequently, it was our aim to evaluate the clinical implications of quantitative biomarkers derived from pre-therapy ^18^F-FDG-PET/CT in a well-characterized retrospective cohort of patients receiving first-line ICI therapy with pembrolizumab in combination with chemotherapy or as monotherapy.

## 2. Materials and Methods

### 2.1. Patients

Eighty-five consecutive patients who had undergone ^18^F-FDG-PET/CT before receiving first-line ICI therapy with pembrolizumab between June 2018 and December 2019 were retrospectively identified, follow-up was accomplished until December 2020. The patient cohort was derived from the institutional NSCLC immunotherapy registry of Kepler University Hospital Linz. The patient registry as well as the present evaluation have been approved by the ethics committees of the federal state of Upper Austria (EK Nr. 1139/2019), the need for patients’ written informed consent was waived. All investigators had full access to the dataset used for this analysis. This study was conducted according to the REporting of studies Conducted using Observational Routinely collected health Data (RECORD) statement [49].

According to institutional standards, patients with PD-L1 expression <50% received a chemo-ICI combination with pembrolizumab and carboplatin/pemetrexed for non-squamous and carboplatin/paclitaxel for squamous histology [10,11]. Chemotherapy was given for four cycles with no further maintenance therapy, pembrolizumab was continued until progression or toxicity. Patients with PD-L1 expression ≥50% could either receive pembrolizumab monotherapy or a combination with platinum-based doublet chemotherapy [48]. Patients were retrospectively followed from first-line ICI therapy initiation on to death or censored at the date of last verified contact. Disease progression and survival were retrospectively assessed by reviewing the relevant medical records, especially imaging studies and death certificates. First-line therapy was defined as the first systemic treatment in stage IV or not otherwise treatable stage III disease, whereas previous therapies in potentially curable stages were not considered. We excluded patients in clinical trials, on ICI/ICI combination therapies and patients, who had previously received ICI for NSCLC or other malignancies.

### 2.2. Image Acquisition Protocol and Analysis

PET/CT imaging was accomplished in the staging process usually two to four weeks before therapy initiation, however, a time span of a maximum of three months was allowed for inclusion if no tumor-specific therapy had been applied in that time. PET/CT scans were performed using a dedicated Siemens Biograph 40 Truepoint PET/CT scanner (Siemens Medical Solutions, Illinois). Patients kept fasting for at least six hours and blood glucose levels were measured before the injection of ^18^F-FDG imaging to ensure that values were below 150 mg/dL. ^18^F-FDG was administered at a dose of 3.7 MBq/kg through a peripheral vein 60 min prior to imaging. Sequential overlapping emission scans of the neck, chest, abdomen, and pelvis were acquired. PET imaging was performed in 3D mode at 3 min per bed position, using the same axial field as the CT scan. We performed image reconstruction using an ordered subset expectation maximization iterative reconstruction algorithm on a 168 × 168-pixel matrix (AW-OSEM, 2 iterations, 8 subsets), followed by post-reconstruction filtering using a Gaussian filter applied at 5.0 mm full width at half maximum. All patients had attenuation-corrected images without intravenous contrast agent application. All PET/CT studies were reviewed by two specialist nuclear medicine physicians, who were blinded to the clinical data. For further analysis of quantitative PET/CT biomarkers, imaging data were transferred to a Hermes Workstation (Hermes Medical Solutions, Stockholm, Sweden). Semiquantitative analysis of ^18^F-FDG tumor uptake was performed with the Affinity Viewer^®^ software tool (Version 1.0, Hermes Medical Solutions, Stockholm, Sweden). In this research, all SUV values were based on body weight. To determine the lesion SUVmax, isocontour regions of interest were semi-automatically drawn over abnormal findings at 50% of the maximum pixel value within the respective lesion, SUVmean and MTV were calculated using an SUV threshold of 41% of SUVmax according to our institution standard for lung cancer evaluation. The volumes of all segmented individual lesions, including the primary tumor as well as all metastatic lesions, were summed to obtain the whole-body MTV for each patient [50]. TLG was calculated as the product of the MTV and the SUVmean within the MTV [51]. BLR and SLR were calculated as the ratio of bone marrow/spleen and liver SUVmax. Bone marrow SUVmax was measured in the vertebral bodies of L1-L5, whereas areas with vertebral fractures and tumors/metastases were omitted. Spleen and liver SUVmax were calculated in a spherical VOI of three cm in the respective organ in an area with physiological morphology in the CT images, excluding, e.g., metastases.

### 2.3. Laboratory Analyses

C-reactive protein (CRP) and LDH were assessed using a Cobas^®^ 8000 modular analyzer (Roche Diagnostics International AG, Rotkreuz, Switzerland), lymphocyte count was analyzed using a Sysmex^®^ XN-3000 hematology analyzer (Sysmex Europe GmbH, Norderstedt, Germany). Expression of PD-L1 in tumor cells was determined using a 22C3 assay for Autostainer Link 48 by Dako (Agilent Technologies, Santa Clara, CA, USA), a negative PD-L1 status was defined as membranous staining on <1% of viable tumor cells.

### 2.4. Response Assessment

Radiological response to ICI therapy was routinely assessed every six to nine weeks by a CT scan of the chest and the upper abdomen using iodinated contrast medium unless contraindicated. Re-staging could be preponed due to suspected disease progression and additional imaging modalities such as cerebral magnetic resonance tomography could be conducted according to the treating clinician’s judgment. The response was graded according to the Response Evaluation Criteria in Solid Tumors (RECIST), version 1.1 [52]. Disease control rate (DCR) was defined as patients with complete/partial remission (CR/PR) or stable disease (SD) versus those with progressive disease (PD). Patients who died before the first scheduled CT re-staging (*n* = 11) were counted as PD.

### 2.5. Statistics

Statistical analyses were performed using R (R: A Language and Environment for Statistical Computing; Version 3.6.0). Progression-free and overall survival (PFS, OS) for all patients and in specified subgroups were evaluated using Kaplan–Meier analyses, results were expressed as the median in months (M) with 95% confidence interval (CI). The Kaplan–Meier curves were statistically compared using the log-rank test, whereas a *p*-value < 0.05 was regarded statistically significant. Uni- and multivariate models for PFS and OS were accomplished using Cox regression analyses. For MTV and BLR, cut-off values for PFS and OS were calculated using graphical analysis in quartiles and maximally selected rank (MSR) statistics. Clinical variables included in the multivariate models were age (</≥70 years), sex, smoking history (</≥5 pack years), histological subtype (adeno-, squamous-cell carcinoma), ECOG (0,1/≥2), CRP (</≥0.5 mg/dL), LDH (</≥250 U/L), lymphocyte count (</≥1 G/L), and PD-L1 expression (positive/negative). Patients with any of the relevant variables missing were excluded from the multivariate regression models.

## 3. Results

Quantitative PET/CT biomarkers were available in all 85 patients, patient disposition and baseline characteristics are presented in Figure 1 and Table 1. At the time of analysis, 68 subjects had shown tumor progression, while 53 had died, resulting in a PFS of 5 M (4–8) and an OS of 14 M (7–18). Median follow-up time from baseline PET/CT acquisition to the end of observation was 12 M (10–14), and the median time from baseline PET/CT to ICI therapy initiation was 23 days (21–29).

Using a Cox regression model including all quantitative PET/CT biomarkers, univariate analyses for PFS showed significance for MTV (*p* < 0.001), TLG (*p* = 0.002) and BLR (*p* = 0.046), while stepwise multivariate selection revealed only MTV (*p* < 0.001) as significant. Similarly for OS, univariate analyses indicated significant interactions for MTV (*p* = 0.001), TLG (*p* = 0.003) and BLR (*p* = 0.003), while multivariate selection again showed significance only for MTV (*p* < 0.001). Due to these results, MTV and BLR were defined as respective tumor- and immunologically related quantitative PET/CT biomarker for subsequent analyses. Using graphical analysis of quartiles and MSR calculation, the optimum cut-off values for MTV and BLR regarding PFS were determined at value of 70 mL and 1.06, respectively.

PFS and OS differed significantly according to the defined MTV and BLR subgroups as shown in Table 2 and Figure 2, whereas lower MTV and lower BLR were associated with a more favorable prognosis.

The best radiological response according to RECIST for MTV and BLR subgroups is visualized in Table 3. Response rates differed significantly between patients with MTV ≤/> 70 mL; DCR was 81% for MTV ≤ 70 mL vs. 53% for MTV > 70 mL (*p* = 0.007).

In an exploratory approach, we estimated the prognostic power of MTV and BLR in the context of other, more established, patient- and tumor-related prognostic biomarkers in uni- and multivariate regression models for PFS and OS. As shown in Table 4, univariate analyses for PFS indicated a significant interaction for LDH ≥ 250 U/L, presence of brain metastases, and MTV > 70 mL, while the multivariate model showed significance only for MTV. Concerning OS, univariate analyses revealed an ECOG performance status ≥2, LDH > 250 U/L, MTV > 70 mL, and BLR > 1.06 as significant, while ICI-monotherapy, LDH ≥ 250 U/L, PD-L1 positivity, and BLR > 1.06 had significant implications on the multivariate model.

To identify clinically relevant patient collectives defined by quantitative PET/CT biomarkers, four subgroups with MTV ≤/> 70 mL and BLR ≤/> 1.06 were analyzed for PFS and OS, respectively. As shown in Table 5 and Figure 3, the subgroup with MTV > 70 mL and BLR > 1.06 had considerably reduced PFS/OS, while patients with MTV > 70 mL showed a prognostic benefit if their BLR concomitantly was ≤1.06. Two exemplary cases of patients with high/low MTV and BLR, respectively, are shown in Figure 4.

Figure 4 ^18^F-FDG-PET/CT studies of two cases demonstrating the relationship between MTV and BLR. The patient in (a) presented with a large, right upper lobe tumor with central necrosis (MTV 184 mL) and had a high BLR of 1.73. The patient in (b) had a small intrapulmonary tumor recurrence and local bone metastases one year after initial chemotherapy, resection, and radiotherapy of a stage III Pancoast tumor of the right upper lobe. MTV was 8.1 mL, BLR was 0.73. Boxes indicate the lumbar vertebral bodies; arrows indicate the localization of the primary tumor.

In a subgroup analysis among patients with PD-L1 ≥ 50% (*n* = 29), individuals having received ICI-monotherapy had inferior PFS and OS as compared to chemotherapy-ICI combination, regardless of MTV. For MTV ≤ 70 mL, median PFS, and OS were not reached in the chemo-ICI group and amounted to 4 M (1–6) and 14 M (1-/) in the mono-ICI group, respectively. Patients with MTV > 70 mL had a median PFS and OS of 3 M (2–10) and 6 M (3–18) with chemo-ICI and 2.5 M (1–7) and 3 M (1-/) with mono-ICI.

## 4. Discussion

Our analyses indicate that among the quantitative PET/CT variables evaluated, MTV was the most relevant tumor-related prognostic biomarker in first-line ICI-treated NSCLC. Patients with lower MTV ≤ 70 mL had not only significantly longer PFS and OS, but also a significantly higher radiological response and disease control rate as compared to patients with a higher metabolic tumor burden. Additionally, bone marrow metabolism as assessed by BLR may have the potential to differentiate between favorable and adverse prognoses especially in those patients with higher MTV. In uni- and multivariate analyses for PFS and OS, both MTV and BLR showed hazard ratios comparable to traditional prognostic factors such as ECOG performance status or the presence of brain metastases.

From a clinical point of view, it is not surprising that metabolically active tumor burden as measured by MTV turned out as the most relevant tumor-related quantitative PET/CT biomarker in our cohort. Similar observations have been reported for various disease stages of NSCLC, using PET/CT or conventional CT imaging for both the determination of baseline tumor burden and response to ICI [21,23,30,53,54]. Concerning first-line treatment using mono-ICI therapy, Dall’Olio et al. recently reported an MTV ≥ 75 cm^3^ as a biomarker of poor prognosis in a cohort of 34 pembrolizumab-treated NSCLC patients with PD-L1 expression ≥50%, with an OS of 4.7 M (0.3–9.1), while median OS was not reached in patients with MTV < 75 m^3^ [22]. These results are similar to our findings with an MTV threshold calculated at 70 mL, but with a better OS of 10 M (5–15) in our reported MTV > 70 mL group, which may be due to the addition of chemotherapy in the majority of patients. Seban et al. evaluated a cohort of 63 patients in the same therapeutic setting and identified MTV > 84 cm^3^ and SUVmean > 10.1 as significant predictors of long-term benefit, PFS and OS [25]. Similarly, Yamaguchi et al. reported on 48 patients treated with first-line pembrolizumab for NSCLC with PD-L1 ≥ 50% and identified MTV as significant uni- and multivariate prognostic determinant [26]. In a cohort with 42 out of 57 NSCLC patients being treated with first-line ICI, Polverari et al. found associations of higher MTV and TLG with radiological disease progression [24]. All these studies consistently show very similar findings as reported in our cohort, especially concerning the major prognostic implications of MTV. However, in our patient collective, the vast majority received chemotherapy-ICI combination treatment, which reflects the current clinical practice in a considerably larger patient population as compared to the discussed evaluations of mono-ICI in NSCLC with PD-L1 ≥ 50%. Our findings regarding BLR could have an additional impact on patient management, since lower BLR may identify patients with a better treatment response despite higher tumor burden. This resembles previous reports in cohorts of NSCLC and cutaneous melanoma patients, where the combination of MTV with DLNR [30,31], TLG with DLNR [27], as well as of MTV with BLR [37] provided similar prognostic information.

Reviewing these results, the question arises, how such biomarker information derived from PET/CT could benefit clinical decision making on the individual patient’s level. Currently, for NSCLC patients with higher baseline MTV and BLR and without specific molecular targets, alternative first-line treatment options next to mono-ICI or chemotherapy-ICI combination are not available. Still, our findings have several implications on daily clinical practice: First, we suggest that PET/CT should be performed at baseline in all advanced NSCLC patients receiving ICI treatment. Given the current scarcity of prognostic biomarkers in these patients, our results and previous evaluations clearly suggest that biomarkers derived from PET/CT have prognostic relevance. Second, in line with other authors [22], we propose that patients with a high tumor burden as depicted by MTV should receive chemo-ICI combination rather than mono-ICI therapy. Third, patients at risk for early progression as identified by our reported PET/CT-derived biomarkers should be monitored more closely during initial therapy. Treating clinicians should timely ensure the availability of a complete panel of currently targetable genetic tumor alterations needed for second-line treatment decisions, such as the presence of a KRAS p.G12C mutation [55]. Such patients could also benefit from participation in clinical trials on substances aiming at enhancing the anti-cancer activity of existing (chemo-)immunotherapy agents, e.g., Canakinumab or Tiragolumab [56,57]. Moreover, novel molecular imaging tracers and “theranostic” substances currently under development for different tumor entities could provide new incentives in that field [4,58,59,60,61,62].

Our reported analysis has inherent limitations, but also strengths that should be addressed: The limited sample size, the single-center- and retrospective study design warrant further larger-scale and prospective trials in that field. Still, it represents a considerable portion of especially chemotherapy-ICI combination-treated NSCLC patients. This allows an insight into a patient collective of currently high clinical relevance, as most metastatic NSCLC patients receive such combination treatment at the moment. Another limitation is the current lack of standardized methods for the evaluation of quantitative PET/CT biomarkers, which limits the comparability between studies, although we sought to use similar approaches as suggested by previously published evaluations. In the future, methodological inconsistencies between centers, e.g., in the calculation of MTV, could be overcome by increased application of machine learning- and artificial intelligence-based segmentation and biomarker calculation algorithms, as well as by the rapidly evolving field of radiomics [24,63,64,65]. General limitations of PET/CT in thoracic malignancies naturally also apply to our reported findings: These include its limited sensitivity in small lesions or tumors with low cell density and reduced FDG avidity such as in bronchoalveolar carcinoma and misinterpretation of uptake in benign lesions caused by inflammatory processes [17]. In addition, we used standard RECIST rather than immunotherapy-specific iRECIST for radiological response assessment [66], and follow-up PET/CT was not conducted. Bone marrow hypermetabolism has been repeatedly reported as a prognostic factor in various tumor entities including NSCLC [37,38,39,40,43,44,45]; however, occasionally it can be difficult to differentiate between immunologically related lymphoid tissue hyperactivity and tumor- or trauma-associated bone abnormalities. Although we sought to exclude metastatic lesions of fractures within the region of interest in the lumbar vertebral bodies, diffuse metastatic bone marrow infiltration ultimately cannot be ruled out, as bone marrow biopsies are usually not assessed in stage IV NSCLC patients due to the lack of clinical consequences.

## 5. Conclusions

Quantitative baseline PET/CT biomarkers in ICI-treated advanced NSCLC patients can provide essential prognostic biomarker information, both concerning metabolic tumor characteristics, but also reflecting the immune system. The combination of high MTV and BLR identifies a clinically highly relevant group of patients with a poor prognosis that warrants intensified diagnostic and therapeutic efforts by the clinician as well as future research activity concerning additional treatment options.

## Figures and Tables

**Figure 1 cancers-13-06096-f001:**
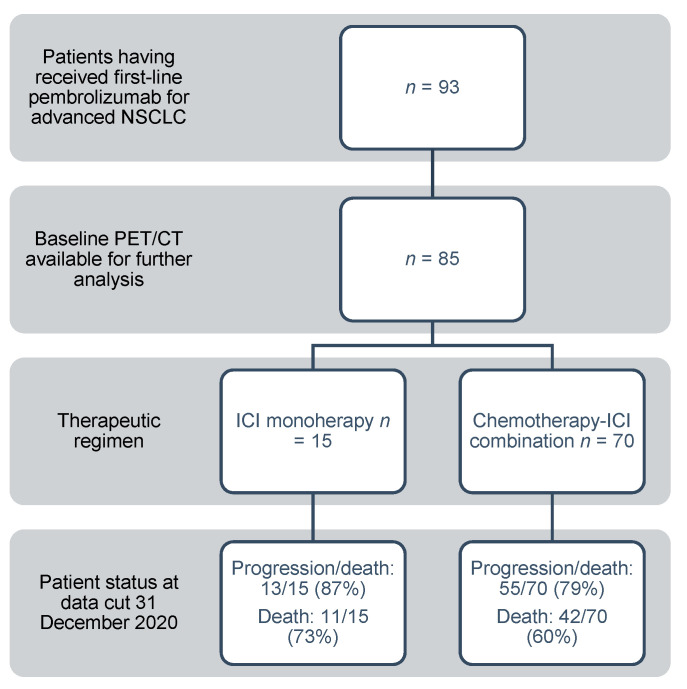
Patient disposition. NSCLC = non-small cell lung cancer, PET/CT = positron-emission tomography/computed tomography, ICI = immune checkpoint inhibitor.

**Figure 2 cancers-13-06096-f002:**
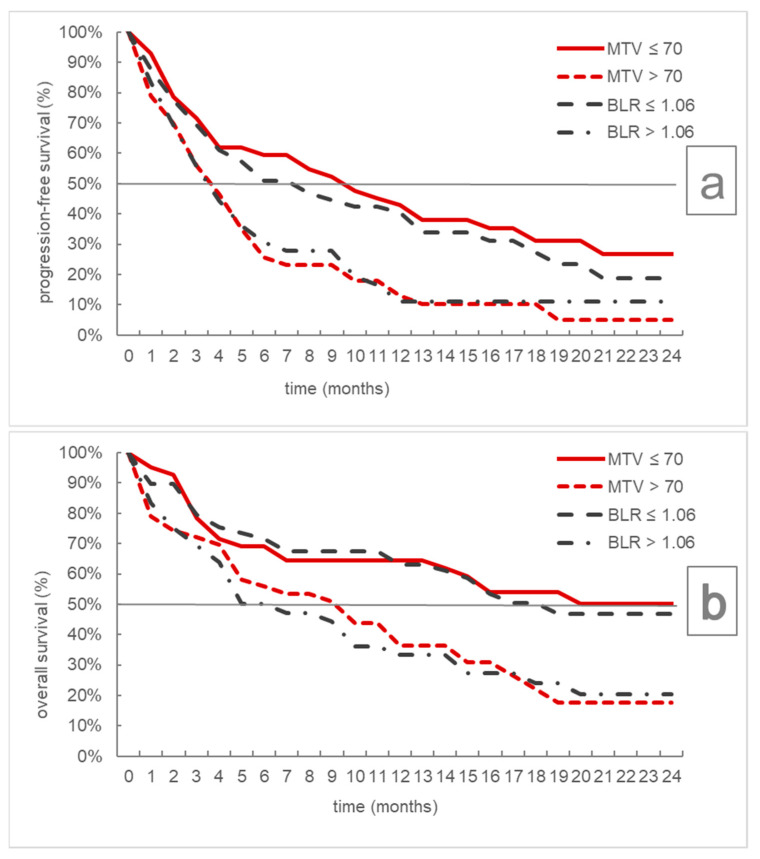
Kaplan–Meier curves for progression-free (**a**) and overall survival (**b**) according to MTV and BLR subgroups. MTV = metabolic tumor volume, BLR = bone marrow to liver ratio.

**Figure 3 cancers-13-06096-f003:**
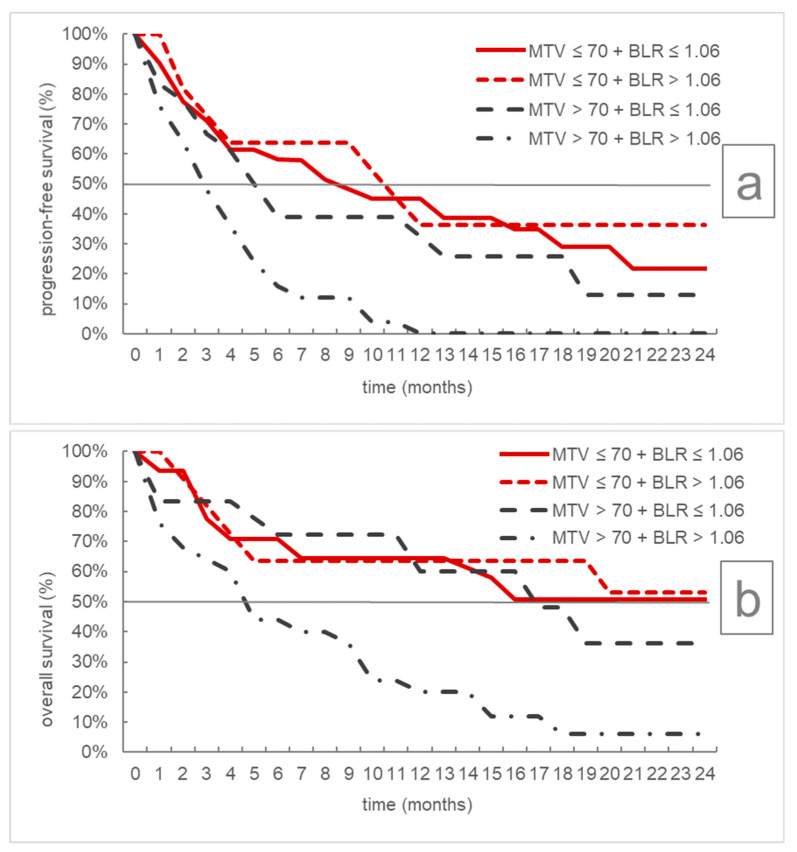
Kaplan–Meier curves for progression-free (**a**) and overall survival (**b**) according to combined MTV and BLR subgroups. MTV = metabolic tumor volume, BLR = bone marrow to liver ratio.

**Figure 4 cancers-13-06096-f004:**
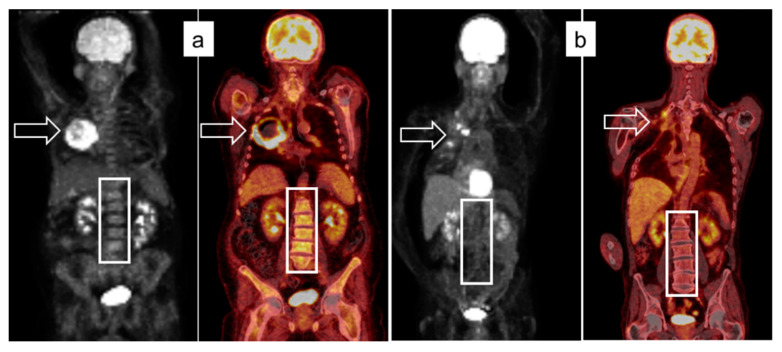
^18^F-FDG-PET/CT studies of two cases demonstrating the relationship between MTV and BLR. The patient in (**a**) presented with a large, right upper lobe tumor with central necrosis (MTV 184 mL) and had a high BLR of 1.73. The patient in (**b**) had a small intrapulmonary tumor recurrence and local bone metastases one year after initial chemotherapy, resection and radiotherapy of a stage III pancoast tumor of the right upper lobe. MTV was 8.1 mL, BLR was 0.73. Boxes indicate the lumbar vertebral bodies; arrows indicate the localization of the primary tumor.

**Table 1 cancers-13-06096-t001:** Baseline characteristics. ECOG = Eastern Cooperative Oncology Group, SD = standard deviation, ICI = Immune checkpoint inhibitor, IQR = interquartile range, NSCLC = non-small cell lung cancer, PD-L1 = programmed death-ligand 1, PET/CT = positron-emission tomography/computed tomography, SUV = standardized uptake value.

Patient Characteristics	
Median age (range; years)	64 (38–81)
Male sex (*n*, %)	56 (66)
ECOG (*n*, %)	
0	42 (49)
1	27 (32)
2+	16 (19)
Presence of brain metastases (*n*, %)	32 (37.6)
Smoking history ≥ 5 pack years (*n*, %)	79 (89.4)
Pack years (mean, SD)	44.5 (24.3)
**Therapy Characteristics**	
ICI monotherapy (n, %)	15 (17.6)
Median number of mono-ICI cycles (IQR)	3 (2.5)
Chemotherapy-ICI combination (*n*, %)	70 (82.4)
Median number of chemotherapy-ICI cycles (IQR)	4 (2)
Median number of mono-ICI maintenance cycles (IQR)	2.5 (8)
**Tumor Characteristics**	
Histological subtype (*n*, %)	
Adenocarcinoma	62 (73)
Squamous-cell carcinoma	22 (27)
NSCLC not otherwise specified	1 (1)
Positive PD-L1 status (*n*, %)	49 (58)
PD-L1 expression (*n*, %)	
Not available	5 (6)
<1%	31 (36)
1–49%	20 (24)
≥50%	29 (34)
**Blood Biomarkers (mean, SD)**	
C-reactive protein (mg/dL)	3.2 (5.3)
Lactate dehydrogenase (U/L)	331.2 (612)
Lymphocyte count (G/L)	1.3 (0.78)
**PET/CT Biomarkers (mean, SD)**	
SUVmax	16 (6.7)
SUVmean	7 (1.8)
Total metabolic tumor volume (mL)	121.6 (145.9)
Total lesion glycolysis	888.6 (1184.3)
Bone marrow to liver ratio	1.04 (0.27)
Spleen to liver ratio	0.81 (0.12)

**Table 2 cancers-13-06096-t002:** Median progression-free and overall survival according to MTV and BLR cutoff values. CI = confidence interval, MTV = metabolic tumor volume, BLR = bone marrow to liver ratio.

	Progression-Free Survival			Overall Survival		
	Median	95% CI	*p*	Median	95% CI	*p*
MTV ≤ 70 mL	10	4–16	0.001	Not reached	7-/	0.004
MTV > 70 mL	4	3–5		10	5–15	
BLR ≤ 1.06	8	4–13	0.034	19	12-/	0.005
BLR > 1.06	4	3–6		6	4–12	

**Table 3 cancers-13-06096-t003:** Radiological best response and disease control rate according to RECIST. RECIST = Response Evaluation Criteria in Solid Tumors, CR = complete remission, PR = partial remission, SD = stable disease, PD = progressive disease, MTV = metabolic tumor volume, BLR = bone marrow to liver ratio.

			RECIST Best Response				Disease Control Rate	
	Cut-Off	*n*	CR, PR	SD	PD	*p*	CR, PR, SD	*p*
MTV	≤70 mL	42	22 (52)	12 (29)	8 (19)	0.026	34 (81)	0.007
	>70 mL	43	14 (33)	9 (21)	20 (46)		23 (53)	
BLR	≤1.06	49	23 (47)	12 (24)	14 (29)	0.536	35 (71)	0.317
	>1.06	36	13 (36)	9 (25)	14 (39)		22 (61)	

**Table 4 cancers-13-06096-t004:** Uni- and multivariate analyses for progression-free and overall survival in all patients with full dataset available (*n* = 73). *p*-value indicates the statistical significance of the hazard ratio. HR = hazard ratio, CI = confidence interval, ICI = immune checkpoint inhibitor, ECOG = Eastern Cooperative Oncology Group, SD = standard deviation, IQR = interquartile range, NSCLC = non-small cell lung cancer, LDH = lactate dehydrogenase, CRP = C-reactive protein, PD-L1 = programmed death-ligand 1, MTV = metabolic tumor volume, BLR = bone marrow to liver ratio.

	Univariate		Multivariate		Univariate		Multivariate	
	HR (95% CI)	*p*	HR (95% CI)	*p*	HR (95% CI)	*p*	HR (95% CI)	*p*
	Progression-Free Survival				Overall Survival			
ICI-monotherapy vs. chemotherapy-ICI combination	1.33 (0.70–0.52)	0.378			1.50 (0.74–3.04)	0.258	4.01 (1.63–9.87)	0.003
Sex (male vs. female)	1.13 (0.66–1.95)	0.654			1.01 (0.55–1.84)	0.985		
Age (>70 vs. ≤70 years)	1.18 (0.67–2.07)	0.567			1.16 (0.60–2.23)	0.666		
ECOG (2+ vs. 0,1)	1.64 (0.88–3.03)	0.117			2.20 (1.11–4.38)	0.025		
Histology (squamous cell vs. adenocarcinoma)	1.25 (0.69–2.24)	0.464			1.53 (0.80–2.93)	0.199		
>5 packyears (yes vs. no)	0.55 (0.24–1.30)	0.174			0.62 (0.25–1.57)	0.315		
LDH (>250 vs. ≤250 U/L)	1.80 (1.05–3.07)	0.032			2.22 (1.23–4.00)	0.008	4.34 (2.02–9.33)	<0.001
CRP (>0.5 vs. ≤0.5 mg/dL)	1.27 (0.64–2.51)	0.492			1.52 (0.68–3.40)	0.306		
PD-L1 (pos. vs. neg)	1.22 (0.73–2.05)	0.457			1.29 (0.72–2.31)	0.384	3.55 (1.54–8.14)	0.026
Lymphocyte count (>1 vs. ≤1 G/L)	1.16 (0.68–1.98)	0.578			1.03 (0.57–1.87)	0.914		
Presence of brain metastases (yes vs. no)	1.70 (1.02–2.84)	0.043			1.45 (0.85–2.59)	0.170		
MTV (>70 vs. ≤70 mL)	1.90 (1.12–3.23)	0.017	1.90 (1.12–3.23)	0.015	1.88 (1.03–3.42)	0.040		
BLR (>1.06 vs. ≤1.06)	1.63 (0.98–2.72)	0.061			2.10 (1.18–3.74)	0.012	2.09 (1.16–3.75)	0.014

**Table 5 cancers-13-06096-t005:** Median progression-free and overall survival according to combined MTV and BLR subgroups values. CI = confidence interval, MTV = metabolic tumor volume, BLR = bone marrow to liver ratio.

		Progression-Free Survival			Overall Survival		
	*n*	Median	95% CI	*p*	Median	95% CI	*p*
MTV ≤ 70 mL + BLR ≤ 1.06	31	9	4–18	<0.001	Not reached	7-/	<0.001
MTV ≤ 70 mL + BLR > 1.06	11	11	2-/-		Not reached	3-/	
MTV > 70 mL + BLR ≤ 1.06	18	5.5	3–13		17	6-/	
MTV > 70 mL + BLR > 1.06	25	3	2–5		5	2–10	

## Data Availability

As mandated by the Ethics Committee of Upper Austria, publication or dissemination of any possibly identifiable patient data from the present registry is prohibited. The dataset used for the present analyses contains very detailed and thus possibly identifiable patient data, therefore, publication of the full database is not possible. However, upon reasonable request to the authors and if permitted by the Ethics Committee of Upper Austria in an amendment to the study protocol, anonymized data can under certain circumstances be shared.

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
