# Peer review of "First-Line Pembrolizumab Mono- or Combination Therapy of Non-Small Cell Lung Cancer: Baseline Metabolic Biomarkers Predict Outcomes"

_cancers, 2021, doi:10.3390/cancers13236096_

Round 1

Reviewer 1 Report

GENERAL COMMENTS Good study. Well written. The authors investigated the prognostic value of FDG-PET features before first-line chemo-immunotherapy or immunotherapy alone of patients with advanced NSCLC. This innovative work could lay the foundations to identify strong predictive/prognostic biomarkers for tumor response to upfront immunotherapy in combination with chemotherapy based on systemic inflammation, tumor uptake and metabolic tumor burden on initial 18F-FDG PET/CT scan.

However, several points must be explained and/or corrected.

SPECIFIC COMMENTS

Simple summary
- Line 22: I would say “prognostic” rather than “predictive”. This should be corrected in the entire manuscript accordingly.
- Line 24: I think you are talking about TMTV (total metabolic tumor volume) rather than MTV?

Abstract:
- Please precise the median follow-up with CI 95%.

Introduction
- The first paragraph is a bit long, and information could probably be condensed without much loss.
-Line 58-59: The end of the sentence “although by far not all respond to such treatment” is not clear to me. Please rephrase.
-Line 72: Another paper published by Cancers should be cited on the combination of neutrophils-derived indices and FDG-PET biomarkers in a specific cohort of advanced NSCLC patients with ≥50% PD-L1 receiving frontline pembrolizumab (PMID: 32785166).
- Line 76-77: “Although bone marrow hypermetabolism is a known prognostic factor in resected or chemo(-radio)therapy treated NSCLC, implications for ICI therapy of advanced NSCLC have not yet been reported”: I do not agree with you. You should read the following paper, recently published in Lung Cancer (PMID: 34311344).
- The subsequent analysis of lymphoid organs using FDG-PET is not really well introduced. It may be good to have an additional short explanation of why the high inflammation process is associated with a poor prognosis (high activities in the bone marrow and/or in the spleen) in a wide range of tumors (melanoma, breast…), using appropriate citations.

Methods
- Please add the RECORD statement.
- Line 138-139: “To determine the SUVmax, irregular isocontour regions of interest were drawn over abnormal findings at 50% of maximum pixel value within the lesion.” Can you explain the reason why you choose this methodology?
- The use of RECIST 1.1 criteria rather than iRECIST should be discussed in the limitation section.
- For PD-L1 expression, can you precise the criteria for positivity (tumor cells, immune cells, both?, > 1% ?). - How did the authors deal with missing data? Results - The results are clear and interesting, worth further investigation, and may benefit the management and treatment decision of advanced NSCLC patients in the first-line setting.
- Please add the median follow-up period with 95%CI.

Discussion
- Line 295-296: “Also, concomitant radiotherapy of the primary tumor or of metastatic lesions could further enhance outcomes in such cases”. I don’t understand this sentence and also why it is mentioned here. The abscopal effect is in my opinion an altogether different matter.
- Line 308-309: I would remove this sentence or rephrase it because many papers are about to be published on this issue (advanced NSCLC patients undergoing chemo-immunotherapy).
- I fully agree with the fact that bone marrow metabolism is not well understood at this point. However, several papers studying other tumors (e.g. gynaecological: PMID 33500424, or melanoma: cf. pubmed) suggest the cross-talk between tumor cells, immune cells in the microenvironnement and immunosuppressive cells in the blood, bone marrow or spleen (as well as macrophages, MDSCs, monocytes). I really think that a small paragraph should be added for pathophysiological rationale.

Figures and tables - Figures and tables are clear and relevant.
- Please add PET images (MIP only?) for patients with high/low (T)MTV and high/low BLR.

References
- The list is up to date but not perfectly adapted since several references should be added (see above).

Reviewer 2 Report

General remarks

This is an interesting study about the use of baseline FDG PET-CT in patients with NSCLC who receive either pembrolizumab monotherapy or a combination of pembrolizumab and chemotherapy within first line treatment. The results confirm the impact of baseline metabolic tumor volume (MTV) as a prognostic biomarker for prediction of response to therapy. Although this result is not new, it is of clinical value. Furthermore, they found that bone marrow/liver ratio (BLR) has also a prognostic impact but not as a single parameter.

The main limitation is the retrospective character of the study as well as the lack of follow-up PET-CT studies.

Specific comments

  1. The title should be changed in “….pembrolizumab mono – or combination therapy of…..”
  2. The affiliations are not correct. No. 3 should be no. 4. Please check.
  3. Abstract, ln 34: ….bone marrow- / or spleen to….
  4. Pg 3, Image Acquisition Protocol and Analysis, ln 139: the calculation of MTV is not clear. Do the authors include also metastatic lesions in this calculation or did they measure only the MTV of the primary tumor? What about NSCLC tumors with low FDG uptake? Please comment on this and discuss it.
  5. The manually calculation of MTV is a problem, it strongly depends on the window and the type of VOI´s used. The authors should discuss these problems and mention the potential use of AI-based segmentation algorithms in future for a more standardized quantification.
  6. Pg 3, ln 129: 128 x 128 pixel matrix is relatively low, why this?
  7. Pg 4, response assessment: why did the authors not use at least an FDG PET-CT for follow-up and response assessment?
  8. Table 4: the values for MTV in the multivariate analysis for PFS are not correct, please check. What is the p-value?
  9. Pg 10, ln 244: “…in the chemo-ICI group and….” And should be replaced by with? Please check.
  10. The discussion part should be revised as mentioned above.
